# Adverse Childhood Experiences among Adults in North Carolina, USA: Influences on Risk Factors for Poor Health across the Lifespan and Intergenerational Implications

**DOI:** 10.3390/ijerph17228548

**Published:** 2020-11-18

**Authors:** Adam Hege, Erin Bouldin, Manan Roy, Maggie Bennett, Peyton Attaway, Kellie Reed-Ashcraft

**Affiliations:** 1Public Health Program, Department of Health and Exercise Science, Appalachian State University, Boone, NC 28608, USA; bouldinel@appstate.edu; 2Department of Nutrition and Healthcare Management, Appalachian State University, Boone, NC 28608, USA; roym1@appstate.edu; 3Department of Epidemiology, Mailman School of Public Health, Columbia University, New York, NY 10032, USA; margaretben07@gmail.com; 4Research Triangle Institute, Raleigh, NC 27616, USA; pattaway@rti.org; 5Department of Social Work, Appalachian State University, Boone, NC 28608, USA; ashcraftkb@appstate.edu

**Keywords:** adverse childhood experiences (ACEs), social determinants of health, health disparities, Behavioral Risk Factor Surveillance System (BRFSS), North Carolina

## Abstract

Adverse childhood experiences (ACEs) are a critical determinant and predictor of health across the lifespan. The Appalachian region of the United States, particularly the central and southern portions, experiences worse health outcomes when compared to the rest of the nation. The current research sought to understand the cross-sectional relationships between ACEs, social determinants of health and other health risk factors in one southcentral Appalachian state. Researchers used the 2012 and 2014 North Carolina Behavioral Risk Factor Surveillance System (BRFSS) for analyses. An indicator variable of Appalachian county (*n* = 29) was used to make comparisons against non-Appalachian counties (*n* = 71). Analyses further examined the prevalence of ACEs in households with and without children across Appalachian and non-Appalachian regions, and the effects of experiencing four or more ACEs on health risk factors. There were no statistically significant differences between Appalachian and non-Appalachian counties in the prevalence of ACEs. However, compared with adults in households without children, those with children reported a higher percentage of ACEs. Reporting four or more ACEs was associated with higher prevalence of smoking (prevalence ratio [PR] = 1.56), heavy alcohol consumption (PR = 1.69), overweight/obesity (PR = 1.07), frequent mental distress (PR = 2.45), and food insecurity (PR = 1.58) in adjusted models and with fair or poor health only outside Appalachia (PR = 1.65). Residence in an Appalachian county was independently associated with higher prevalence of food insecurity (PR = 1.13). Developing programs and implementing policies aimed at reducing the impact of ACEs could improve social determinants of health, thereby helping to reduce health disparities.

## 1. Introduction

Health disparities, often rooted in social determinants of health, are a pressing public health challenge in the United States as researchers and numerous policy makers strive for health equity [1,2,3,4]. Within this scope, in recent years, adverse childhood experiences (ACEs) have emerged in the public health research and practice fields as a critical factor linked with health disparities across the lifespan [5,6,7,8]. Typically, ACEs are measured among adults and refer to one having a childhood embroiled with such challenges as emotional, physical, or sexual abuse; substance abuse or mental illness in their household, or incarceration of a family member [9]. Research has specifically identified that those experiencing four or more ACEs are the most prone to major health conditions later in their life [6,7,10]. In addition, for those adults having experienced ACEs as a child, it can affect their children and their health as well.

ACEs are more likely to be reported among individuals residing in communities and neighborhoods where there are increased levels of social and economic stress experienced by families [11,12]. From an intergenerational framework, this can lead to ACEs playing a significant role in children encountering increased adverse health outcomes through social determinants of health [12,13,14,15]. In their study of 10 U.S. states and the District of Columbia, Metzler and colleagues [13] revealed that those adults experiencing four or more ACEs were more likely to have not completed a high school education, had an income below the poverty level and had experienced extended periods of unemployment. A recent study also found that those adults who had experienced four or more ACEs was correlated with being a single parent and either having Medicare/Medicaid/Children’s Health Insurance Program (CHIP) for insurance coverage or no health insurance [16]. Each of these life circumstances can significantly impact life opportunities for both adult parents and their children.

Research has shown that ACEs can also contribute to one’s parenting behaviors and practices, which in turn have significant effects on the health and wellbeing of children in the household [17,18,19]. In a previous study, researchers reported that mothers who had experienced childhood trauma were more likely to practice an authoritarian style of parenting [20], while Newcomb and Long [21] found specifically that childhood sexual abuse was associated with aggressiveness as a parent. More specifically, parents with authoritarian or aggressive tendencies are more likely to utilize increased physical forms of punishment or engage in physical or sexual abuse and maltreatment of their children. Using data from the National Survey of Children’s Health, Crouch’s research team [22] reported that parents experiencing increased stress levels were more than three times more likely to have children that would experience four or more ACEs by the age of 18. Concerningly, another study [19] found that children with parents who had experienced four or more ACEs were more likely to be hyperactive and have an emotional disturbance diagnosis, when compared to children with parents experiencing no ACEs. Meanwhile, in their recent study, Wang and colleagues [23] found that parental exposure to ACEs and parenting stressors served as a mediating role for adolescent delinquency and behavioral problems among youth. Therefore, it is critical to further understand the prevalence of ACEs among adults with children currently living in the household as it can have serious consequences for the experiences of the children.

Much research has documented the role of geographic location as it pertains to social determinants of health and health disparities, and the inequities found among rural areas and recent research has shown that rural areas are more prone to ACEs [12,24,25,26]. The largely rural Appalachia region of the U.S., which spans 13 states and more than 400 counties, has been identified as one of the geographic locations encountering a vast array of health and social challenges, culminating in vast health disparities when compared to the rest of the country [27,28]. Some of these consequences can be traced back to the largely rural context of the region; rural America, as a whole, has greater risks for obesity, mental health disorders, substance abuse, heart disease, maternal and child health, and tobacco use [29]. It is also important to examine differences within the region; for instance, Hendryx and colleagues [30] related that portions of southern and central Appalachia are most disadvantaged. Specific to ACEs, previous research has shown that southern and predominantly rural states generally have higher prevalence rates [31]. It is highly plausible that ACEs play a significant role in shaping intergenerational social and health challenges found in the region.

Based on the literature depicting the importance of ACEs to health across the lifespan as well as the social determinants of health, health disparities, and poor health outcomes found across Appalachia, we sought to examine relationships between these aspects for the state of North Carolina. The Appalachian counties in western North Carolina are found within southern Appalachia. Specifically, we made use of data from North Carolina’s Behavioral Risk Factor Surveillance System (BRFSS) to make comparisons between the 29 Appalachian counties and the other 71 non-Appalachian counties in the state and hypothesized that the respondents residing in Appalachian counties would have more experiences with ACEs and unfavorable social determinants of health and poorer health outcomes. We also hypothesized that given the differences in social determinants between the two regions that the impact of ACEs may be different for people residing within the Appalachian region of the state compared to the rest of the state. Finally, we aimed to compare the ACEs experiences among adults with and without children in their households in both the Appalachian and non-Appalachian regions of the state to measure the extent to which current children’s experiences may differ by region.

## 2. Materials and Methods

In this study, we made use of the 2012 and 2014 North Carolina BRFSS data and analyses were performed in fall 2019. The BRFSS is a representative telephone (landline and cellular) survey of adults age 18 and older designed to provide public health surveillance data based on self-reported information [32]. Each year, the BRFSS samples an independent set of adults; therefore, the respondents included in the two years were not the same individuals and this is a cross-sectional study. The BRFSS, coordinated by the Centers for Disease Control and Prevention (CDC), is conducted annually in all US states and territories and includes core questions along with optional modules and state-added questions. The public dataset does not include county of residence; therefore, we obtained a dataset from the North Carolina BRFSS coordinator with an indicator variable for residence in the 29 counties that constitute Appalachia based on the Appalachian Regional Commission’s definition [33]. The BRFSS includes an Adverse Childhood Experiences optional module and a single item on food insecurity in the Social Context optional module. In 2012, the ACE module questions were included on all three versions of North Carolina’s BRFSS, while in 2014 the module was asked only of respondents to version 1 of the landline and cellphone survey (representing two of the five versions offered that year). These were the latest years the ACEs module was included on North Carolina’s BRFSS. This study was classified as exempt by the Institutional Review Board at Appalachian State University (IRB #18–0071).

### 2.1. Adverse Childhood Experiences (ACEs)

With the ACEs module, participants were asked a series of eleven questions on the following topics: living with someone who (1) had serious mental illness, (2) misused alcohol, (3) used illegal drugs or misused prescription drugs, or (4) had been incarcerated; having parents or adults in the home who (5) were separated or divorced, (6) physically assaulted one another, (7) physically hurt the respondent, or (8) verbally hurt the respondent; having someone older or an adult (9) touch the respondent sexually or (10) force the respondent to touch them sexually or (11) have sex with them. Respondents reported whether they had experienced the adverse event before the age of 18 or how frequently they had experienced it (never, once, or more than once), depending on the item. We classified respondents as having experienced an ACE if they had ever experienced it (i.e., “ever” or “once” or “more than once”). The questions and response options appear in full in Appendix A.

We summed the number of ACEs reported. Based on previous studies that reported four or more ACEs predicted significantly higher risk of poor health outcomes as an adult [6,7,10], we created a dichotomous indicator for experiencing 0–3 versus four or more ACEs. We also created three subscales—household dysfunction, emotional or physical abuse, and sexual abuse—based on a study by Ford and colleagues [34]. We classified respondents as experiencing each domain of ACE if they scored a 1 or higher on relevant items. Household dysfunction included items 1–5, emotional or physical abuse included items 6–8, and sexual abuse included items 9–11. We excluded 2267 respondents (432 in Appalachia and 1835 outside Appalachia) because they did not answer at least one ACE question. To evaluate the impact of this exclusion on our results, we conducted a sensitivity analysis in which we included all respondents who answered more than half (at least six) of the ACEs items.

### 2.2. Social Determinants of Health

We used the following demographic and household characteristic variables broadly as social determinants of health: age; sex; race/ethnicity; annual household income; highest educational attainment; if respondents currently had children in their household; and food insecurity. We categorized race/ethnicity as: non-Hispanic white; non-Hispanic African American; Hispanic; and other/multiracial. Household income included: <$15,000; $15,000–24,999; $25,000–49,999; $50,000–74,999; and ≥$75,000. Education included: less than high school; high school/GED; technical school/some college; and college degree or higher. Because we included age, sex and educational attainment in regression models, we excluded respondents who were missing any of these variables.

Food insecurity was only measured in the 2012 data, and the question was, “How often in the past 12 months would you say you were worried or stressed about having enough money to buy nutritious meals?” Response selections included: always; usually; sometimes; rarely; and never. We considered those reporting always, usually, or sometimes to be food insecure. This single item in the BRFSS has been validated and determined by the U.S. Department of Agriculture (USDA) to be a simplified version of their standardized measures for food security [35].

### 2.3. Health Status, Behaviors & Risk Factors

On the BRFSS, respondents are asked to rate their health as: excellent; very good; good; fair; or poor. We categorized the responses as “excellent, very good or good” and “fair or poor”. For smoking, we made use of the question, “Do you now smoke cigarettes every day, some days, or not at all?” We considered a smoker to be one that responded with “every day” or “some days”. We also classified people who reported they had smoked less than 100 cigarettes in their lifetime as non-smokers.

Regarding alcohol consumption, among the respondents who reported having at least one drink during the past 30 days, participants were further asked, “On the days when you drank, about how many drinks did you drink on average?” Heavy alcohol users were considered men who have more than two drinks a day and women who have more than one drink per day; this is a CDC-calculated variable in the BRFSS dataset. For body mass index (BMI) we used the three categories of: not overweight or obese (≤24.99); overweight (25.0–29.99); and obese (≥30.0).

Mental distress was measured by the following question: “Now thinking about your mental health, which includes stress, depression, and problems with emotions, for how many days during the past 30 days was your mental health not good?” We considered frequent mental distress to be 14 or more days of poor mental health [36].

### 2.4. Statistical Analysis

The final analytic sample included 13,050 adults; 2506 lived in Appalachia and 10,544 lived outside Appalachia. In the full state sample (weighted to represent the state), 29% were age 18–34, 17% were age 35–44, 18% were age 45–54, 16% were age 55–64, and 19% were age 65 or older. About 53% were women, 68% were non-Hispanic white, 20% were non-Hispanic black, 5% were non-Hispanic from another racial group, and 7.5% were Hispanic. Forty-three percent had a high school degree or less, 32% had completed some college, and 25% had earned a college degree or higher. Twelve percent of the sample lived in households with an annual income below $15,000 and 26% lived in a household with an annual income of $75,000 or higher. One-third of the sample had a child in the household. We calculated the weighted proportion of respondents who experienced each individual ACE, each of the three categories of ACEs, and also the percentages of respondents with 0, 1, 2, 3, and 4 or more ACEs in Appalachian and non-Appalachian regions of the state by whether or not there were children in the household. We focused on the issue of children in the household due to the aforementioned role that experiences with ACEs can shape parenting behaviors and practices among adults and often lead to intergenerational challenges with ACEs. All prevalence comparisons between Appalachian and non-Appalachian regions of the state were made using chi-square tests. We describe the sample using weighted percentages and compare the characteristics of people with 0–3 ACEs to those with four or more ACEs within and across regions using chi-square tests, ignoring any missing values in a list-wise manner.

We used weighted and adjusted log-binomial regression models to estimate the associations between having four or more ACEs (compared to 0–3 ACEs) and (1) fair or poor health, (2) current smoking, (3) heavy alcohol consumption, (4) overweight or obese (BMI ≥ 25.0), (5) frequent mental distress, and (6) food insecurity. Give than these data are cross-sectional and that the outcomes were common and therefore the odds ratio would likely not approximate the relative risk, we estimated the prevalence ratio using a log-binomial regression model (i.e., a generalized linear model with a binomial family and log link) [37]. We tested for effect modification by residence in Appalachia by including an interaction term between Appalachian residence and having an ACE score of four or higher.

We adjusted all models for age and age squared, sex, and educational categories to account for differences in demographics and social determinants between Appalachia and the rest of the state. Sample sizes vary across regression models depending on the number of missing values for each outcome. We re-ran these six models using respondents who had answered at least six ACE items as sensitivity analysis (Appendix A). For all analyses, we used the appropriate weight variable in the BRFSS file via survey (svy) commands with subpopulation statements in Stata 13.1, consistent with BRFSS guidance [38].

## 3. Results

Nearly 60 percent of respondents reported at least one ACE and one in six (16.6%) reported four or more ACEs (Table 1). In Appalachia, 56.8% of respondents reporting any household dysfunction had at least one child under the age of 18, compared to 41.1% of respondents reporting any household dysfunction without children, whereas outside of Appalachia the results were 56% and 44% respectively. For any emotional or physical abuse, findings revealed that 35.6% of those with a child in the household in Appalachia had the experience while 33.9% with no children had the experience; outside of Appalachia, the numbers were slightly higher at 39.3% and 34.3%. Regarding sexual abuse, the numbers were consistently around 10%, with the highest prevalence being 13.1% among adults outside of Appalachia who had a child in the household. 17.2% of respondents in Appalachia, with children in the household (15.4% for those without children), reported four or more ACEs, while 21.1% outside of Appalachia (14.4% for those without children) reported four or more ACEs. Table 1 provides the results, including confidence intervals. The prevalence of ACEs was similar across the state regardless of Appalachian or non-Appalachian residence. Household dysfunction, particularly parental separation or divorce, and emotional abuse were most prevalent.

Across all regions, people with 0–3 ACEs and 4–11 ACEs generally had similar demographic and household characteristics and health (Table 2). Within Appalachia, respondents with 4 or more ACEs were younger and more likely to be current smokers. They also more frequently reported having poor mental health and being food insecure. Compared to people outside Appalachia with the same category of ACEs, people within Appalachia tended to be older and more often non-Hispanic white. Table 2 further provides the *p*-values and confidence intervals.

We found a statistically significant interaction between health status and Appalachian region. Within Appalachia, an ACE score of four or more was not associated with fair or poor general health status, while outside Appalachia it was (Table 3). In all other models there was no evidence of an interaction by region, so we report model results adjusting for residence in Appalachia (Table 4). After adjusting for age, sex, and education, all other outcomes evaluated were more common among people with four or more ACEs across all North Carolina residents. Having four or more ACEs was associated with a slight increased prevalence of overweight/obesity (PR = 1.07), while current smoking, heavy alcohol consumption, and food insecurity were associated with a 50–60% higher prevalence (PR = 1.53; PR = 1.60; and PR = 1.61, respectively), and frequent mental distress was associated with a 2.5-fold higher prevalence (PR = 2.52). Living in Appalachia was only associated with higher food insecurity (PR = 1.13).

## 4. Discussion

More than half of the North Carolina adults (59.6%) who responded to the survey have experienced at least one ACE. This finding is consistent with the original Felitti [5] study in which researchers reported 64 percent of respondents (*n* = 17,337) had experienced at least one ACE. It was concerning, in our study, that those with children were more likely to report ACEs. There is an increased risk for the children currently in the household to have similar experiences, which can lead to intergenerational effects [11,12,13], particularly in rural environments [15], and North Carolina is largely rural. Within Appalachia, adults reporting four or more ACEs were most commonly aged 45–54. This could signal a pathway between ACEs and the ongoing crises of “diseases of despair” among this age group and deserves further attention [39,40]. Those adults residing in Appalachia who reported four or more ACEs had nearly a double (34.2% vs. 18.8%) smoking, a triple (23.5% vs. 9.4%) frequent mental distress, and a double (45.9% vs. 22.7%) food insecurity prevalence. The magnitude of associations between health behaviors and outcomes generally was similar to that reported in earlier studies. The prevalence ratios reported here are smaller than odds ratios reported elsewhere, likely because, as noted above, the odds ratio is a poor approximation of the relative risk when outcomes are common. Contrary to our hypothesis, there were no significant differences in the prevalence of ACEs by Appalachian region. Residence in Appalachia did not generally confer added risk of poor health behaviors or outcomes after accounting for ACEs and demographic/social determinant of health differences, with the exception of food insecurity. Finally, while we did find evidence of effect modification for one outcome (general health status), it was not in the direction anticipated. For people living in Appalachia, there was no impact of ACEs on general health, while for people outside Appalachia, experiencing four or more ACEs increased the risk of experiencing poor health.

Our results generally align with three large multi-state U.S. studies as well as previous work from North Carolina. Specifically, as with our results, Metzler [13], Merrick [41], and Sonu [42] found that four or more ACEs was associated with higher odds of household poverty and lower educational attainment. However, our results were not as pronounced as previous studies. Further, Merrick [41] and Sonu [42] also reported, much like with our findings, that ACEs were more prevalent among younger and middle-age groups. It is highly plausible that the younger and middle age groups are more likely to self-report ACEs and that their experiences (changes in society, social norms, etc.) with ACEs could have a more significant impact. Using Wave II data from the original ACE study, researchers also found increased odds for poor mental health outcomes, including heavy alcohol use and frequent mental distress, among those with higher ACE scores [43]. In a large global meta-analysis [7], increased odds (OR = 2.20) for heavy alcohol use and depression (OR = 4.40), were found among those reporting four or more ACEs. These previous results, much like the findings from our study, show an important need for more attention directed at the longer-term impacts of ACEs on mental health outcomes and alcohol abuse. Few previously published studies have explored the relationship between ACEs and food insecurity, but Sun and colleagues [44] also found that both poor mental health and ACEs were independently associated with food insecurity. Specific to North Carolina residents, Roy and colleagues [45] recently reported that each additional ACE experienced was associated with a 13–21 percent increase in the odds of being food insecure. More research needs to be directed at understanding the linkages between ACEs and food insecurity challenges.

Interestingly, we found that the relationship between ACEs and general self-rated health differed by region of the state. Among adults living in Appalachia, there was no relationship between ACEs and fair or poor health, while in the rest of the state, experiencing four or more ACEs increased the prevalence of fair or poor health substantially. Previous studies have found that ACEs or maltreatment in childhood are associated with poorer self-rated health or lower health-related quality of life in adulthood [7,46]. It is not clear why adults in Appalachia would not have a similar experience; it is possible that unique challenges (i.e., poverty, lack of health care access, attitudes toward help and the mistrust of institutions) to health in the region override the impact of ACEs. It could also be that residents of Appalachia are more resilient as it pertains to ACEs or that the social and cultural norms of “pick yourself up by your bootstraps” largely found across Appalachia could diminish the chances of respondents giving an accurate measure of their health. A recent study [47] suggested that rural children develop more resilience due to having protective factors such as more engagement in their community (school, churches) and are more likely to have mentors and the capacity to develop nurturing relationships with adults outside of the home. Much like our findings, however, a previous study [48] also found this result in which Appalachian adults self-report a better health status than what the data would indicate. Research should explore this phenomenon in relation to ACEs across the region through qualitative research methods.

While this study did not find significant differences between Appalachia counties and the remainder of the state, the uniqueness of North Carolina should be noted and utilized in future research. When exploring the state, the ninth most populous in the U.S., and its three sub-regions, the majority of the western and eastern portions of the state are rural, whereas the central portion is largely urban. Not surprisingly, the eastern and western portions fare much worse in terms of social determinants of health and health outcomes [49]. Future research that examines ACEs and the relationships with factors for poor health should seek to consider the rural-urban contextual differences. In addition to the rural-urban contrasts, racial disparities should be explored; specific to North Carolina, the western rural portions of the state are largely white, whereas the eastern portion is much more diverse in terms of race and ethnicity. Institutionalized racism should also be considered in terms of its effects on trauma and adverse childhood experiences (i.e., institutional racism should be considered as an adverse childhood experience). Qualitative research could help to elucidate more of the full picture that goes along with the quantitative evidence.

While these data and the findings are from well before the ongoing COVID-19 pandemic, it is critical that the context is provided. Leading experts and scholars have begun to warn about the effects of the current pandemic on exacerbating ACEs and creating new forms of trauma experienced by children and their families [50,51,52]. There is no doubt that the pandemic has created new forms of stress that are often out of our control, and there is concern that many well-intentioned efforts to mitigate the transmission of COVID-19 could have unintended negative consequences [53]. As Campbell [54] emphasized, it is imperative that communities strengthen and create new and innovative partnerships in simultaneously addressing the ramifications of COVID-19 and the effects on families and children. Researchers investigating both the COVID-19 pandemic and ACEs should pay close attention to this relationship as they conduct their investigations.

The findings from the current study further supports the growing evidence that states such as North Carolina, which comprise the Appalachia region and are largely rural, are in urgent need of a more comprehensive, upstream and systems-level approach to improving health across the lifespan. With the expanding knowledge around factors leading to ACEs, as well as the effects, research and interventions should take a transgenerational, life course and transdisciplinary approach [13,15,55,56]. Local communities and states should consider approaches and frameworks like the Building Community Resilience Model [57,58]. The Building Community Resilience Collaborative housed within the Milken Institute School of Public Health at George Washington University is working collaboratively with communities all across the United States to “improve the health of children, families, and communities by fostering engagement between grassroots community services and public and private systems to develop a protective buffer against ACEs occurring in adverse community environments” [59]. Last, and most importantly, policies and systems at the federal and state levels must be improved to better support families to help prevent social challenges that lead to increased adversity. As encouraged by Metzler [13] and McEwen [58], the substantial evidence that we now have related to ACEs suggests that ACEs should be embedded in work related to social determinants of health and points to the increased needs for such social policies as universal home visiting programs, more accessible and improved housing conditions, accessible and affordable healthy food options, increased wages or universal basic income, and universal health care coverage, to name a few [58,60].

### Limitations

This study has several limitations. First, BRFSS data are cross-sectional and the ACEs questions are retrospective in nature, requiring that respondents accurately remember and report events from many years before the survey. Secondly, this study only included respondents from a single state, therefore, the results may not generalize to other states, even within the Appalachian region. Additionally, although BRFSS samples are weighted to be representative of the population, the response rate is somewhat low (41% in North Carolina 2019), though better than many other similar surveys [61]. Telephone administration excludes people without phones; therefore, these estimates may not reflect the experiences of those populations. Finally, a substantial number of respondents did not answer some or all of the eleven ACEs items. While there appeared to be few differences between the people who were included and excluded in terms of demographics and other characteristics, it is possible that this exclusion resulted in a sample that does not represent the ACEs of all North Carolina residents, which may therefore bias our prevalence estimates. However, based on our sensitivity analyses, we do not believe there is substantial bias in the estimates of the relationship between ACEs and health behaviors or outcomes based on missing data.

## 5. Conclusions

This study, similar to previous research, highlights the role of ACEs in health across the lifespan. Although there were no statistically significant differences between Appalachian and non-Appalachian counties of North Carolina in prevalence of ACEs, the findings indicate that a large proportion of adults who have experienced ACEs are currently raising children. While the data presented in the current study is prior to COVID-19, it is apparent that the global pandemic is only exacerbating the risks associated with ACEs, social determinants of health and other risk factors for poor health. Therefore, it is important to prioritize funding and policies designed to provide universal support to all families, particularly as we mitigate all of the various factors involved with the COVID-19 pandemic. To support this re-prioritization, funding multi-method longitudinal research that captures changes in outcomes and costs is warranted. It is not only time for us to continue to build more knowledge around our understanding of ACEs within the U.S. culture and context, but to move forward in supporting the next generation of Americans. The success of our nation is largely dependent on how we support our children and their families in their health and development across the lifespan.

## Figures and Tables

**Table 1 ijerph-17-08548-t001:** Prevalence of ACEs among adults in North Carolina overall, by presence of children in the household, and by region, 2012 & 2014.

ACE Category	All Adults	Adults in Appalachia			Adults Outside Appalachia		
	(N = 13,050)	Households with Children (N = 554)	Households without Children (N = 1950)	*p*-Value ^1^	Households with Children (N = 2798)	Households without Children (N = 7722)	*p*-Value ^1^
	Weighted % (95%CI)	Weighted % (95%CI)	Weighted % (95%CI)		Weighted % (95%CI)	Weighted % (95%CI)	
	Household Dysfunction						
Household mental illness	15.2 (14.1–16.4)	17.7 (13.0–23.7)	14.9 (11.8–18.7)	0.38	16.8 (14.8–19.0)	14.1 (12.6–15.7)	0.04
Household alcohol abuse	22.7 (21.5–24.0)	25.1 (19.0–32.3)	20.1 (17.1–23.4)	0.16	25.5 (22.9–28.2)	21.6 (20.1–23.2)	0.01
Household substance use	10.7 (9.7–11.7)	10.6 (7.4–15.1)	9.5 (7.1–12.5)	0.62	14.3 (12.1–16.8)	9.0 (7.8–10.4)	<0.001
Incarcerated family member	7.3 (6.5–8.1)	9.9 (6.8–14.2)	5.0 (3.7–6.7)	0.004	9.2 (7.6–11.3)	6.6 (5.6–7.7)	0.008
Parental separation or divorce	28.2 (26.8–29.6)	36.5 (30.1–43.5)	23.0 (19.5–27.0)	<0.001	36.3 (33.3–39.4)	24.4 (22.6–26.2)	<0.001
Any household dysfunction	48.1 (46.6–49.6)	56.8 (50.1–63.)	41.1 (37.4–45.0)	<0.001	56.0 (53.0–58.9)	44.7 (42.8–46.7)	<0.001
	Emotional/Physical Abuse						
Household physical violence	16.3 (15.3–17.4)	15.8 (11.9–20.6)	15.6 (12.6–19.2)	0.96	18.9 (16.7–21.2)	15.2 (13.9–16.6)	0.005
Physical abuse	13.9 (12.9–15.0)	14.7 (10.8–19.8)	15.0 (12.0–18.6)	0.93	15.2 (13.3–17.4)	12.8 (11.6–14.1)	0.04
Emotional abuse	28.1 (26.8–29.5)	30.8 (25.0–37.2)	27.2 (23.7–30.9)	0.31	31.5 (28.7–34.5)	26.1 (24.4–27.9)	0.001
Any emotional/physical abuse	35.8 (24.4–37.2)	35.6 (29.5–42.2)	33.9 (30.3–37.7)	0.65	39.3 (36.4–42.3)	34.3 (32.5–36.2)	0.005
	Sexual Abuse						
Touched	9.4 (8.5–10.3)	9.4 (6.5–13.4)	9.0 (6.4–12.5)	0.87	11.2 (9.3–13.3)	8.5 (7.4–9.7)	0.02
Touched adult	7.0 (6.3–7.9)	8.3 (5.4–12.5)	6.4 (4.4–9.1)	0.36	8.9 (7.3–10.8)	6.1 (5.3–7.1)	0.003
Forced sex	4.3 (3.7–5.0)	5.2 (3.1–8.6)	4.7 (2.6–8.2)	0.79	5.0 (3.9–6.3)	3.8 (3.0–4.7)	0.08
Any sexual abuse	11.1 (10.2–12.2)	11.3 (8.0–15.8)	10.3 (7.6–13.8)	0.69	13.1 (11.1–15.3)	10.3 (9.1–11.6)	0.02
	Number of Adverse Childhood Experiences (ACEs)						
0	40.4 (39.0–41.8)	31.4 (25.9–37.6)	47.4 (43.7–51.2)	0.001	34.4 (31.7–37.2)	42.8 (41.0–44.8)	<0.001
1	23.5 (22.3–24.8)	30.2 (23.5–37.8)	21.0 (18.4–23.8)		22.9 (20.4–25.6)	23.8 (22.2–25.5)	
2	11.2 (10.3–12.2)	12.8 (8.9–17.9)	8.6 (7.0–10.5)		12.9 (10.9–15.2)	10.8 (9.6–12.1)	
3	8.3 (7.5–9.2)	8.5 (5.5–13.0)	7.6 (12.1–19.4)		8.7 (7.1–10.7)	8.2 (7.1–9.4)	
4 or more	16.6 (15.4–17.8)	17.2 (12.9–22.4)	15.4 (12.1–19.4)		21.1 (18.7–23.7)	14.4 (13.0–15.9)	

^1^*p*-value compares households with children and households without children in a given region (i.e., within Appalachian or outside Appalachian) based on chi-square test of weighted data. For the number of ACEs, the *p*-value indicates a trend across all categories (i.e., people with children in the household tended to have more ACEs than people without children in the household). ACE: Adverse childhood experience.

**Table 2 ijerph-17-08548-t002:** Characteristics of respondents and ACEs within Appalachia and outside Appalachia, North Carolina, 2012 & 2014.

Variable	Category	Within Appalachia			Outside Appalachia		
		0–3 ACEs (N = 2174)	4–11 ACEs (N = 332)	*p*-Value ^1^	0–3 ACEs (N = 9014)	4–11 ACEs (N = 1530)	*p*-Value ^2^
		Weighted % (95%CI)	Weighted % (95%CI)		Weighted % (95%CI)	Weighted % (95%CI)	
**Demographic & Household Characteristics**							
Age	18–34	25.4 (21.9–29.2)	27.4 (19.7–36.9)	<0.001	27.3 (25.4–29.2)	41.0 (36.6–45.5)	<0.001
	34–44	12.5 (10.3–15.0)	18.1 (12.0–26.2)		17.2 (15.9–18.6)	22.7 (19.5–26.2)	
	45–54	14.5 (12.4–17.0)	35.9 (25.6–47.5)		17.9 (16.6–19.3)	16.7 (14.3–19.5)	
	55–64	19.8 (17.5–22.4)	10.8 (7.5–15.3)		16.3 (15.2–17.4)	13.5 (11.3–16.0)	
	65+	27.1 (24.6–29.7)	7.7 (5.1–11.5)		20.5 (19.4–21.6)	5.9 (4.8–16.0)	
	Missing	0.7	0.2		0.8	0.2	
Sex	Women	51.5 (48.1–55.0)	62.0 (51.3–71.6)	0.07	50.6 (48.8–52.4)	62.4 (58.3–66.4)	0.63
Race/Ethnicity	Non-Hispanic white	79.9 (77.1–82.4)	79.6 (71.1–86.0)	0.90	65.3 (65.6–67.0)	65.6 (61.6–69.3)	<0.001
	Non-Hispanic black/African American	9.4 (7.7–11.3)	12.5 (8.1–18.8)	0.23	22.5 (21.0–24.1)	17.8 (14.9–21.1)	<0.001
	Hispanic	7.5 (5.9–9.5)	NR	--	7.4 (6.5–8.5)	9.0 (7.1–11.3)	0.94
	Other or Multiracial	2.9 (2.0–4.3)	NR	--	4.3 (3.7–5.0)	7.2 (5.4–9.6)	0.07
	Missing	0.3	0.1		0.5	0.5	
Annual Household Income	<$15,000	10.2 (8.3–12.6)	15.4 (10.4–22.0)	0.32	9.4 (8.5–10.4)	14.5 (12.2–17.2)	<0.001
	$15,000–24,999	19.7 (17.0–22.8)	24.0 (15.2–35.8)		15.9 (14.6–17.4)	20.1 (17.1–23.5)	
	$25,000–49,999	26.1 (23.2–29.2)	30.8 (21.4–42.1)		21.7 (20.3–23.2)	23.2 (19.5–27.3)	
	$50,000–74,999	10.8 (8.7–13.3)	10.9 (6.6–17.5)		12.1 (11.0–13.3)	11.2 (8.9–14.1)	
	≥$75,000	16.3 (14.1–18.8)	10.7 (7.1–15.8)		23.5 (22.0–25.1)	18.7 (15.4–22.5)	
	Missing	16.8	8.2		17.3	12.2	
Highest Level of Education	<High school	17.5 (14.6–20.7)	16.7 (9.0–28.8)	0.13	14.1 (12.9–15.4)	17.4 (14.7–20.4)	0.001
	High school or GED	29.9 (26.9–33.0)	32.6 (23.3–43.5)		26.7 (25.1–28.4)	26.6 (23.0–30.6)	
	Technical school or some college	28.9 (25.8–32.3)	37.2 (28.2–47.2)		31.4 (29.7–33.1)	39.1 (34.8–43.5)	
	College degree	23.7 (21.3–26.4)	13.5 (9.6–18.8)		27.8 (26.4–29.3)	16.9 (14.6–19.6)	
Any Children in the Household	Yes	28.5 (25.1–32.1)	31.2 (23.3–40.3)	0.56	32.6 (30.9–34.4)	43.4 (39.2–47.7)	0.04
	Missing	0.1	0.2		0.3	0.4	
**Health Status, Behaviors & Risk Factors**							
Current Health Rating	Excellent, very good, or good	77.7 (74.8–80.4)	75.0 (66.6–81.9)	0.56	83.9 (82.7–85.0)	70.5 (66.3–74.3)	<0.001
	Fair or poor	22.0 (19.3–25.0)	24.2 (17.5–32.6)		15.8 (14.7–17.0)	29.4 (25.6–33.6)	
	Missing	0.3	0.8		0.3	0.1	
Current Smoker	Yes, some days or every day	18.8 (15.9–22.1)	34.2 (25.5–44.2)	<0.001	16.7 (15.3–18.2)	33.2 (29.2–37.4)	0.20
	Missing	0.6	0		0.4	0.02	
Current Heavy Alcohol Use	Men who have >2 drinks per day and women who have >1 drinks per day	4.5 (3.3–6.1)	7.4 (4.2–12.6)	0.11	4.4 (3.7–5.3)	7.3 (5.4–9.7)	0.97
	Missing	2.6	3.2		2.8	1.9	
Current BMI (kg/m^2^)	Not overweight or obese (≤24.9)	31.8 (28.8–34.9)	28.4 (20.8–37.4)	0.63	32.2 (30.5–33.9)	27.7 (24.1–31.7)	0.44
	Overweight (25.0–29.9)	33.1 (30.1–36.4)	35.4 (25.6–46.6)		35.5 (33.8–37.3)	32.6 (28.8–36.6)	
	Obese (≥30.0)	27.8 (24.6–31.4)	32.2 (23.9–41.9)		26.2 (24.7–27.8)	33.8 (29.9–38.0)	
	Missing	7.2	4.0		6.1	5.9	
Current frequent mental distress	≥14 days of poor mental health in past month	9.4 (7.5–11.6)	23.5 (16.2–33.0)	<0.001	8.5 (7.6–9.4)	26.1 (22.3–30.2)	0.40
	Missing	1.4	0.4		1.2	1.9	
Food insecure *2012 data only*	Always, usually, or sometimes were worried or stressed about having enough money to buy nutritious foods in the past year	22.7 (20.1–25.5)	45.9 (38.4–56.6)	<0.001	19.9 (18.7–21.2)	42.8 (39.3–46.4)	0.06
	Missing	0.6	0		0.6	0.1	

NR: Not reported because of an unweighted denominator < 50 or a relative standard error > 30.0. ^1^
*p*-value < 0.01 comparing people with 0–3 ACEs to those with 4–11 ACEs within Appalachia using a chi-square test (weighted data). For ordered variables (age, income, education, BMI), this chi-square test evaluates the trend across all categories rather than comparing individual categories within the variable. ^2^
*p*-value < 0.01 comparing people within Appalachia to those outside Appalachia within the same category of ACEs (i.e., 0–3 ACEs within Appalachia to 0–3 ACEs outside Appalachia or 4+ ACEs within Appalachia to 4+ ACEs outside Appalachia). For ordered variables (age, income, education, BMI), this chi-square test evaluates the trend across all categories rather than comparing individual categories within the variable.

**Table 3 ijerph-17-08548-t003:** Prevalence ratio (PR) of fair or poor health by Appalachian region, among North Carolina residents, 2012 & 2014.

Exposure Variables	Category	Fair or Poor General Health Status ^1^	*p*-Value
		PR (95%CI)	
Within Appalachia (*n* = 2494)			
Number of ACEs	4–11	1.04 (0.72–1.52)	0.83
	0–3	1.0 (Reference group)	
Outside Appalachia (*n* = 10,502)			
Number of ACEs	4–11	1.73 (1.53–1.95)	<0.001
	0–3	1.0 (Reference group)	

^1^ Adjusted for age, age squared, sex, and education category. CI: Confidence interval.

**Table 4 ijerph-17-08548-t004:** Prevalence ratio (PR) of fair or poor health, current smoking, heavy alcohol consumption, overweight or obesity, frequent mental distress, and food insecurity among North Carolina residents, 2012 & 2014.

Exposure Variables	Category	Outcomes				
		Current Smoking ^1^ (*n* = 12,990)	Heavy Alcohol ^1^ (*n* = 12,652)	Over- weight or Obese ^1^ (*n* = 12,255)	Frequent Mental Distress ^1^ (*n* = 12,851)	Food Insecurity ^1,2^ (*n* = 10,134)
		PR (95% CI)	PR (95% CI)	PR (95% CI)	PR (95% CI)	PR (95% CI)
Number of ACEs	4–11	1.53 * (1.34–1.76)	1.60 § (1.17–2.19)	1.07 § (1.02–1.12)	2.52 * (2.10–3.01)	1.61 * (1.46–1.78)
	0–3	1.0 (reference group)				
Region of Residence	In Appalachia	1.10 (0.94–1.30)	1.09 (0.80–1.49)	0.99 (0.96–1.04)	0.97 (0.78–1.20)	1.13 † (1.02–1.25)
	Outside Appalachia	1.0 (reference group)				

^1^ Adjusted for age, age squared, sex, and education category. ^2^ Uses 2012 BRFSS data only. * *p* < 0.001, § *p*-value < 0.01, † *p*-value < 0.05.

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
