# Peer review of "Adverse Childhood Experiences among Adults in North Carolina, USA: Influences on Risk Factors for Poor Health across the Lifespan and Intergenerational Implications"

_ijerph, 2020, doi:10.3390/ijerph17228548_

Round 1
Reviewer 1 Report
Thank you very much for your manuscript, I greatly enjoyed reading it. Overall, I think this paper makes an important contribution by highlighting the need for intervention - especially in the current climate - to break the cycle of intergenerational 'transmission' of the effects of ACEs.
I have made comments on the pdf file, and have highlighted each position in the manuscript in orange. I have also made a few green highlights for very minor grammatical issues.
A few general points to note are that:
- there is no mention of ethics clearance for the study in the manuscript. I assume that although you used an existing dataset, you still applied for (and were granted) ethical approval for your study?
- in the results, it would be informative to report the chi square values, rather than just the cut-points for p-value.
- some misleading language is used, where phrasing implies causality or the use of a variable as the independent variable in the analyses.
- the tables are not very clear at the moment, especially table 3. There are two lots of information there, with two independent sets of headings. This should be split into two separate tables. Statistical significance is not denoted clearly due to the choice of symbols.
Altogether I feel all the points above, and those highlighted in the manuscript, are mostly about presentation and can easily be addressed in a short amount of time. I look forward to reviewing the amended version of the manuscript.

Author Response
Responses to Reviewers:
The authors greatly appreciate the reviewers’ suggestions and comments that have been provided to help strengthen our manuscript. We have sought to address each of the comments provided. Please find the responses below. Within our manuscript, we have also highlighted where the changes have been made and addressed.
Reviewer #1
We have addressed the comments provided and provided our responses to the comments on the manuscript using Track Changes/Comments.
To the specific points raised in the review:
there is no mention of ethics clearance for the study in the manuscript. I assume that although you used an existing dataset, you still applied for (and were granted) ethical approval for your study?
Yes, the study was approved by the Institutional Review Board at Appalachian State University. It was deemed “exempt” and the IRB #18-0071 is now provided in the manuscript.
in the results, it would be informative to report the chi square values, rather than just the cut-points for p-value.
While we understand the reviewer’s comment and suggestion, we do not include the specific p-values (or chi-square values) because it would simply be too overwhelming for the reader.
some misleading language is used, where phrasing implies causality or the use of a variable as the independent variable in the analyses.
We have sought to clean up the language throughout the manuscript.
the tables are not very clear at the moment, especially table 3. There are two lots of information there, with two independent sets of headings. This should be split into two separate tables. Statistical significance is not denoted clearly due to the choice of symbols.
We have split Table 3 into two separate tables – now Table 3 and Table 4.
Reviewer 2 Report
This manuscript investigates the relationship between adverse childhood experiences and health of individuals living in Appalachian and non-Appalachian counties. I have several suggestions, which in my opinion would increase the clarity and presentation of the manuscript.
- Please specify the hypothesis that were tested. What is/are the focus of this study? a) comparing individuals living in Appalachian and non-Appalachian counties; b) finding (childhood) risk factors for poor health in adulthood; c) both; d) other hypothesis/research questions?
- Please include mean and SD of age of the respondents. Since the ACE should be experienced up to the age of 18, the effect of ACE would likely be stronger for younger individuals. Therefore, it is important to specify the mean age (and SD) of the sample.
- When was the ACE measured? Please also specify, whether this construct was measured multiple times.
- If the focus of this study is on the longitudinal ideas/hypothesis, please also provide some information about the attrition analysis in respect of sample drop-out.
- By reducing an 11 scales-variable into a binary variable as conducted for ACE, one would lose some valuable information. Although authors mentioned a conceptual reason behind this procedure, I would suggest authors to analyze ACE as an 11-scales-variable as well and see whether the similar results are also obtained. Furthermore, it seems wise to discuss about this preference procedure a little bit to inform the readers that authors are well aware of the consequences.
- All mentioned ACEs were treated with similar weight. I believe, experiencing parents’ divorce is less problematic for a child’s future health than experiencing (non-consensual?) sexual relationship with adults.
- In the table 3: analysis for general health status is rather different compared to analysis of other variables such as alcohol consumption, smoking, overweight etc. Since there were no hypothesis/research questions specified: what is/are the reasons for this distinct analysis?
Author Response
Responses to Reviewers:
The authors greatly appreciate the reviewers’ suggestions and comments that have been provided to help strengthen our manuscript. We have sought to address each of the comments provided. Please find the responses below. Within our manuscript, we have also highlighted where the changes have been made and addressed.
Reviewer #2
- Please specify the hypothesis that were tested. What is/are the focus of this study? a) comparing individuals living in Appalachian and non-Appalachian counties; b) finding (childhood) risk factors for poor health in adulthood; c) both; d) other hypothesis/research questions?
The authors appreciate this suggestion. As such, the authors have sought to clarify the hypotheses that were tested and provide more context why each of these was included. The hypotheses statements appear at the end of the Introduction section. We have added in more detail to indicate the rationale for testing effect moderation by region.
- Please include mean and SD of age of the respondents. Since the ACE should be experienced up to the age of 18, the effect of ACE would likely be stronger for younger individuals. Therefore, it is important to specify the mean age (and SD) of the sample.
We cannot calculate a true mean because the BRFSS collapses “age” at 80 years, meaning everyone aged 80 or older is coded as “80”. We present the distribution of age groups in Table 1 to provide similar information.
- When was the ACE measured? Please also specify, whether this construct was measured multiple times.
Although we mentioned the cross-sectional nature of the BRFSS in the analysis section and as a limitation of the study, we did not make the study design explicit earlier in the paper. Therefore, we have added it in the Abstract and earlier in the Methods section when discussing the data source, so it is clearer to readers.
- If the focus of this study is on the longitudinal ideas/hypothesis, please also provide some information about the attrition analysis in respect of sample drop-out.
With this being a cross-sectional study, there is no information to provide regarding attrition. We used data collected from the 2012 and 2014 BRFSS, but for each year, it is an independent sample of adults.
- By reducing an 11 scales-variable into a binary variable as conducted for ACE, one would lose some valuable information. Although authors mentioned a conceptual reason behind this procedure, I would suggest authors to analyze ACE as an 11-scales-variable as well and see whether the similar results are also obtained. Furthermore, it seems wise to discuss about this preference procedure a little bit to inform the readers that authors are well aware of the consequences.
This is true, but another limitation of the approach is that there are very few respondents at the very upper end of the ACEs distribution and we would have had to collapse categories either way to run the regression models. We felt it was better to be consistent with previous studies (cited in the manuscript) than to create our own categories based on the distribution in the data.
- All mentioned ACEs were treated with similar weight. I believe, experiencing parents’ divorce is less problematic for a child’s future health than experiencing (non-consensual?) sexual relationship with adults.
While this could be true, the authors could locate no literature to support this in the present study. We were not aware of any comparisons of types of ACEs in the research literature.
- In the table 3: analysis for general health status is rather different compared to analysis of other variables such as alcohol consumption, smoking, overweight etc. Since there were no hypothesis/research questions specified: what is/are the reasons for this distinct analysis?
We tested all models for effect modification and found it on general health status only, which is why the results are stratified for this outcome but not the others. We added a hypothesis statement in the introduction to better set up this analysis and to provide our rationale for testing for effect modification.
Reviewer 3 Report
To investigate the effects of adverse childhood experiences (ACEs) on health risk factors across the lifespan considering geographical and social determinants of health is an interesting and important topic. Considering the data source and the methods of this manuscript, it should be a good paper. However, I have some concerns before the publication:
- In the introduction section, it is unclear about the research gap.
- Since the main conclusions are similar to previous research, I can see the originality of this study.
- I don't think the title matches the content. In my understanding, the authors have mainly examined the associations between ACEs and health risk factors in the adulthood, indicators of social determinants of health were acted as control variables. Thus, how come the subtitle were named as "The association with social determinants of health and risk factors for poor health across the lifespan"?
- Line 245 to line 254 in the discussion section are more like literature review.
- The authous resort to a hazy reasoning about the differed associations between ACEs and SRH between adults living in Appalachia and adults living outside Appalachia. However, their explanation doesn't convince me.
- I don't understand why the authors arrived at a conclusion like " it is important to prioritize funding and policies designed to provide universal support to all families, particularly as we mitigate all of the various factors involved with the COVID-19 pandemic", based on the content of this study.
Author Response
Responses to Reviewers:
The authors greatly appreciate the reviewers’ suggestions and comments that have been provided to help strengthen our manuscript. We have sought to address each of the comments provided. Please find the responses below. Within our manuscript, we have also highlighted where the changes have been made and addressed.
Reviewer #3
- In the introduction section, it is unclear about the research gap.
In the Introduction section, we have added in a couple of statements to highlight the importance of examining ACEs among adults and differentiating those with children in the household versus those without children in the household. ACEs can have serious consequences for the childhood experiences of those children residing with adults with trauma related to ACEs. It can affect the parenting behaviors and practices and the living conditions that children are raised in. Further, as we highlight in the Introduction, the Appalachian region experiences far worse health outcomes and adverse social determinants of health when compared to the rest of the United States. Therefore, it is important to understand if and how this influences the affects of ACEs in states found within the Appalachian region – North Carolina is a state that has nearly 1/3 of its geographic location found within Appalachia and is largely rural.
- Since the main conclusions are similar to previous research, I can see the originality of this study.
As stated with #1, we have provided context to support the research gap and originality of the study. We also added in about making comparisons between adults and ACEs among those with children in their household against those without children.
- I don't think the title matches the content. In my understanding, the authors have mainly examined the associations between ACEs and health risk factors in the adulthood, indicators of social determinants of health were acted as control variables. Thus, how come the subtitle were named as "The association with social determinants of health and risk factors for poor health across the lifespan"?
The authors have modified the title to “Adverse Childhood Experiences among adults in North Carolina, USA: Influences on risk factors for poor health across the lifespan and intergenerational implications”.
We feel that this title change better reflects that the findings have implications for risk factors for poor health across the lifespan for those who have experienced ACEs and that adults with children currently in their household tended to have more experiences with ACEs. The literature and evidence points toward it affecting the current experiences of children in the same household with adults who had more experiences with ACEs - that they will also be much more likely to also experience more ACEs through an intergenerational affect.
- Line 245 to line 254 in the discussion section are more like literature review.
We sought to clean this up some and make more connections with the findings of the current study.
- The authous resort to a hazy reasoning about the differed associations between ACEs and SRH between adults living in Appalachia and adults living outside Appalachia. However, their explanation doesn't convince me.
The authors have added in another statement to explain or question the findings. The authors also recommend that further research examines these findings using qualitative research methodology.
- I don't understand why the authors arrived at a conclusion like " it is important to prioritize funding and policies designed to provide universal support to all families, particularly as we mitigate all of the various factors involved with the COVID-19 pandemic", based on the content of this study.
The authors feel that it is vital that researchers and policymakers are paying close attention to ACEs as we continue to mitigate the COVID-19 pandemic. Mental health and social determinants of health are affecting the lives of people all across the world and in the U.S. and in North Carolina. The U.S. is facing serious challenges in directing appropriate funding and policy directives aimed at assisting families and their children as a result of COVID-19 and the general political dynamics. Our results (even with data prior to COVID-19) show the importance of prioritizing ACEs and policies aimed at building community resilience – COVID-19 is only exacerbating this along the way. We have added a statement in to our conclusions to make the connections.
Round 2
Reviewer 3 Report
I think the majority of my concerns has been addressed.
Author Response
We thank the reviewer for reviewing our manuscript and for providing some great recommendations for the improvement of our paper.